# Extremity Exercise Program in Breast Cancer Survivors Suffering from Chemotherapy-Induced Peripheral Neuropathy: A Feasibility Pilot Study

**DOI:** 10.3390/healthcare10040688

**Published:** 2022-04-06

**Authors:** Chih-Jung Wu, Ya-Ning Chan, Li-Yu Yen, Yun-Hen Chen, Chyi Lo, Ling-Ming Tseng, Ya-Jung Wang

**Affiliations:** 1Department of Hematology and Oncology, China Medical University Hospital, No. 2, Yude Rd., Taichung 404332, Taiwan; elvaamy@gmail.com; 2School of Nursing, University of North Carolina at Chapel Hill, 120 N. Medical Dr. Carrington Hall #CB 7460, Chapel Hill, NC 27599-7460, USA; chanyn@live.unc.edu; 3Research Nurse, Department of Internal Medicine, National Taiwan University Hospital, Rm. 607, 6F., Laboratory Building, No. 1, Changde St., Taipei 10048, Taiwan; liyu820121wp@gmail.com; 4Department of Nursing, Taipei Veterans General Hospital, No. 201, Sec. 2, Shih, Pai Rd., Taipei 11217, Taiwan; yhchen12@vghtpe.gov.tw; 5School of Nursing, China Medical University, No. 100, Sec. 1, Jingmao Rd., Taichung 406040, Taiwan; chyilo@mail.cmu.edu.tw; 6Department of Surgery, Taipei Veterans General Hospital, No. 201, Sec. 2, Shih, Pai Rd., Taipei 11217, Taiwan; lmtseng@vghtpe.gov.tw; 7Department of Nursing, DaYeh University, No. 168, University Rd., Dacun, Changhua 51591, Taiwan

**Keywords:** extremity exercise program, ten skilled hand exercise, Buerger-Allen exercise, chemotherapy induced peripheral neuropathy (CIPN), breast cancer survivors

## Abstract

Objectives: To evaluate the feasibility of implementation of an extremity exercise program and to examine its preliminary effects in breast cancer survivors suffering from chemotherapy-induced peripheral neuropathy (CIPN). Sample & Setting: Thirteen breast cancer survivors from one hospital in northern Taiwan. Methods and Variables: A single group with repeated measures, and a quasi-experimental design. The intervention program was a four week, home-based extremity exercise program that was comprised of 10 skilled hand exercises and Buerger-Allen exercises. The Total Neuropathy Scale (clinical version), Functional Assessment of Cancer Therapy/Gynecologic Oncology Group, Neurotoxicity (13-Item Version), Identification Pain Questionnaire, and pain Visual Analogue Scale were used to measure CIPN before exercise (T1), during (T2~T4), and after exercise (T5). Qualitative data were also collected at each time point. Data were analyzed by using descriptive statistics, generalized estimating equations, and directed content analysis. Results: None of the participants reported adverse events during the study period. The extremity exercise program significantly improved patient-reported CIPN after intervention at T4 or T5 but was insignificant on clinician-assessed CIPN. The qualitative data of participant experience indicated that this program is feasible and easy to follow. Conclusion: The extremity exercise program is feasible but needs to increase the sample size and prolong the intervention period for confirmation.

## 1. Introduction

Breast cancer is the most common cancer in women worldwide, with an estimated 2.3 million new cases in 2021 [1]. According to a 2020 report by the Health Promotion Administration, Ministry of Health and Welfare of Taiwan, more than 56% of newly diagnosed breast cancer patients received chemotherapy in Taiwan in 2018 [2]. However, some commonly used breast cancer chemotherapy drugs, such as platinum or taxane, have been identified as causing chemotherapy-induced peripheral neuropathy (CIPN) [3]. CIPN affects 11–80% of breast cancer patients across the chemotherapy continuum [4,5]. CIPN is reported after the initiation of chemotherapy and may last for years following the completion of chemotherapy [4,5]. Cancer survivors who are suffering from CIPN may experience sensory, motor, and autonomic symptoms, such as numbness, paresthesias, neuropathic pain, or muscle weakness [6]. With these discomforts, cancer survivors may encounter emotional distress [7], functional decline [8], sleep disturbances [9], and even a lower quality of life (QOL) [10]. Considering the long-term and negative impacts of CIPN, it is crucial to develop effective interventions to manage neuropathic symptoms.

To manage CIPN, researchers have identified various underlying causal mechanisms of CIPN, such as mitochondrial and vascular dysfunction, oxidative stress, DNA damage, and neuroinflammation [3], which have allowed other researchers to further develop and test various non-pharmacological interventions, such as exercise, cryotherapy, acupuncture, or scrambler therapy [11,12]. Specifically, Tofthagen et al. proposed that exercise can reduce neuropathy symptoms and improve balance, muscle strength, and QOL by increasing oxygen and glucose delivery to mitochondria and supplying blood to peripheral nerves. However, the effectiveness of exercise interventions for neuropathic symptoms is still being debated [13].

According to the American Society of Clinical Oncology’s CIPN guidelines, after reviewing a total of four randomized controlled trials (RCTs), the strength of the evidence regarding exercise interventions and the benefits of exercise interventions in treating and managing CIPN is weak [11]. In contrast, Tanay et al. reviewed 13 intervention studies and concluded that exercise interventions have potential benefits for CIPN [14]. Although mixed results were identified, several studies included in the review showed significant improvements in pain, neuropathy symptoms, balance, muscle strength, and QOL in cancer survivors after exercise programs [14]. These studies paid particular attention to aerobic, muscle strength, and balance training [14]. To our knowledge, however, no research has focused specifically on the extremities. Therefore, based on the conceptual model of increasing blood supply to peripheral nerves to reduce neuropathy symptoms [13], our exercise program focused on increasing peripheral blood flow using extremity exercises.

Our extremity exercise program comprises the “Ten Skilled Hand Exercise” (TSHE) and the Buerger-Allen Exercise (BAE). The TSHE is part of traditional Chinese medicine and focuses on clapping or hitting acupoints over the upper extremities to dilate peripheral blood vessels and stimulate overall blood circulation [15]. An expert in traditional Chinese medicine (coauthor Chyi Lo) has verified this process. In addition, the BAE focuses on position changing of the lower extremities to increase peripheral blood flow to the heart [16]. The BAE has also been found to increase perfusion over the lower extremities and to improve neuropathy symptoms among individuals with diabetes-induced peripheral neuropathy [17]. Moreover, the Oncology Nursing Society has emphasized developing interventions to manage symptoms to improve survivors’ outcomes as one of the research priorities between 2019 and 2022 [18]. Therefore, this study aims to evaluate the feasibility of implementing an extremity exercise program for breast cancer survivors with CIPN and examine the preliminary effectiveness of this program on neuropathic symptoms.

## 2. Materials and Methods

### 2.1. Design and Participants

This was a single group, repeated measures, quasi-experimental study. The outcome measure was conducted five times, at the baseline (T1, prior to the intervention) and at the end of each week (T2~T5). Purposive sampling was employed. The inclusion criteria included (1) aged 18 years or order; (2) being diagnosed with breast cancer; (3) having completed chemotherapy; and (4) complaining of CIPN symptoms with an assessment result of Total Neuropathy Scale–clinical version (TNSc) score above 0. We excluded those who were (1) diagnosed with brain, spine, or spinal cord metastasis, (2) spinal cord diseases, (3) a second type of cancer, or (4) psychiatric illness. The institutional review board of Taipei Veterans General Hospital (2016-05-009B) approved this study. The authors declare that the procedures followed were in accordance with the regulations of the responsible Clinical Research Ethics Committee and in accordance with those of the World Medical Association and the Helsinki Declaration.

Patients were referred by the cooperating attending physician in a medical center located in northern Taiwan between September 2016 and March 2018. After a potential eligible patient was referred, a research assistant (RA) called the patient to explain the study and provide sufficient time for the patient to consider participating. Once the patient agreed to participate, the RA scheduled a time with the patient to meet in person to sign the written informed consent before undertaking any study activity. After obtaining the informed consent, the RA assessed the TNSc to confirm the survivor’s eligibility. Those had a TNSc > 0 filled in the questionnaire and watched an extremity exercise program video on YouTube. After watching the video, sufficient time was provided for participants to ask any questions related to the intervention. A weekly log for recording exercise adherence and adverse events was distributed to participants. Lastly, participants’ contact information was obtained and a time for a follow-up meeting was arranged. The research team’s contact information was given to participants in case they had questions. Before each follow-up meeting, the RA called participants to confirm the meeting’s time and place.

The qualitative inquiry method seeks to understand the feasibility of the extremity exercise program of patients utilizing face-to-face interviews along with filed notes to collect data. The semi-structured questions in one-on-one interviews lasted 20–30 min. Interviews were conducted with patients at home, in restaurants, or at the hospital. Guidelines for the interview included the following: (1) How do you feel about the program? (2) Did you encounter obstacles when performing the exercises? (3) Do you think the exercise program is acceptable or practical? (4) Did you gain benefits or experience setbacks during the program? (5) Is there anything else you would like to share with us or with other patients regarding the program?

### 2.2. Extremity Exercise Program

This is a four-week extremity exercise program comprising the video-taped procedure of TSHE and BAE in Mandarin. The TSHE is a 10-action hand exercise program lasting 2.5 to 5 min. Actions 1 through 9 involve clapping or hitting the hands 36 times in each acupoint to stimulate the various meridians. The steps are as follows: (1) flat clapping two purlicues, (2) clapping two lateral palms, (3) hitting two wrists, (4) clapping two purlicues (one against the other at a 90-degree angle), (5) hitting opened fingers (one against the other), (6) the left fist hitting right palm, (7) the right fist hitting the left palm, (8) clapping the two backs of the hands, and (9) using thumbs and index fingers to rub the earlobes from top to bottom. The tenth action involves rubbing the two palms six times until they are warm, placing the palms on the eyelids, and moving the eyeballs from left to right for six cycles; this must be done six times.

Participants had to perform the TSHE four times per day (after breakfast, lunch, and dinner, and before retiring) and to finish one cycle of the above 10 actions in under five minutes. The BAE includes three steps: (1) lie down on the bed, raise the legs between 45 and 60 degrees for one minute supported by a pillow, and maintain that position for 3 min once balance has been achieved; (2) sit up at the bedside, move the foot from extension to flexion, and then change position from the posterior varus to valgus for 3 min; and (3) lie down and warm the legs using a blanket for 3 min. Participants had to repeat the above three steps for three cycles twice a day immediately after waking up and before retiring.

### 2.3. Sample Size Calculation

Sim found that a sample size of 12 in a Tai Chi feasibility study for breast cancer survivors was able to detect a minimum important difference within-group over time but would not allow for a robust causal inference with a single group analysis [19]. Therefore, this study recruited a minimum of 12 participants to examine the feasibility of the intervention.

### 2.4. Outcome Measures

Outcomes included safety, feasibility, clinician-assessed CIPN (Total Neuropathy Scale–clinical version, TNSc), and patient-reported CIPN (Functional Assessment of Cancer Therapy/Gynecologic Oncology Group–Neurotoxicity 13 Item Version [NTX-13], Identification Pain Questionnaire [ID-pain], and pain Visual Analogue Scale [pain VAS]).

#### 2.4.1. Safety and Feasibility

Safety was examined via the occurrence of patient-reported adverse effects. The feasibility was examined via participants’ retention rate, adherence rate to the exercise program, and experience. The retention rate was defined as the percentage of participants who completed the four-week extremity program. The adherence rate was defined as the percentage of upper and lower extremity exercise sections completed by participants each week. These data were collected using the weekly log and the semi-structured interviews. Specifically, the weekly log was given to the participants to record any adverse events, sections of the TSHE and BAE that were completed daily, and participants’ feelings during the program.

#### 2.4.2. Clinician-Assessed CIPN

The TNSc was utilized to assess the objective severity of CIPN. The TNSc consists of seven items including sensory symptoms, motor symptoms, autonomic symptoms, pin sensitivity, vibration sensitivity, strength, and deep tendon reflex (DTR). Each item scores from 0 to 4. The total score ranges from 0–28. The higher score indicates worse CIPN [20]. The scale has tested its convergent validity, discriminant validity, inter- (0.86) and intra-rater (0.86–0.88) reliability in the cancer population [20]. We included sensory symptoms, pin sensitivity, and vibration sensitivity as sensory CIPN; and motor symptoms, strength, and DTR as motor CIPN [20,21].

#### 2.4.3. Participant-Reported CIPN

The NTX-13 is a 13-item measure used to assess subjective neurotoxicity which included sensory CIPN, motor CIPN, as well as auditory and cold sensitivity. Each item scores from 0 to 4. The total score ranges between 0 and 52; 0~4 for each item. A higher the NTX-13 score indicates less CIPN symptoms. The internal consistency, content validity, and concurrent validity have been tested with sensitivity to change over time in breast cancer survivors [22,23].

The ID-pan has six items. It was used to measure subjective neuropathic pain. The total score of ID pain is between −1 and 5. A score of 2 or higher indicates neuropathic pain. “Yes” is scored as 1 and −1 for item 1~5 and item 6, respectively. “No” is scored as 0 for all items. The ID pain has been tested for its intra-class correlation (0.72–0.74) and k statistic (0.74–0.69) [24,25].

The pain VAS is a one-item measure which scores from 0 to 100. A higher pain VAS score indicates more severe pain. It is known to be valid and to be sensitive to measures of pain intensity and was used to measure subjective peripheral pain in this study [26]. The reliability, construct validity, and concurrent validity of the VAS has been tested in cancer studies with an intra class correlation of 0.38~0.89 [27,28].

## 3. Data Analysis

SPSS 22 software was used to analyze the quantitative data. Mean, standard deviation, frequency, and percentage were used to describe the distribution of sample characteristics, clinician-assessed CIPN, patient-reported CIPN, retention rate, and adherence rate. The generalized estimating equation was used to test the preliminary effectiveness of the exercise program on CIPN over time. Qualitative data were saved as a Word document and analyzed by the primary investigator utilizing directed content analysis. Analysis of participant experience was guided by a framework focus on acceptability, demand, implementation, and practicality, the necessary focus of any feasibility study [29]. The researchers who analyzed the data have more than 15 years research or clinical experiences related to breast cancer care and CIPN management, and possess high sensitivity to the meaning of the participants’ experiences. The results of the directed content analysis were passed to research team members for examination, discussion, and feedback.

## 4. Results

### 4.1. Recruitment and Follow-Up

A total of 138 breast cancer survivors were referred, and 89 verbally agreed to participate and were screened for eligibility. The enrollment rate was 64.5%. After screening with the TNSc, 13 survivors (14.8%) were deemed eligible, finished the baseline measure and began the intervention. Two participants, one with severe pain in the lower extremities and a second with a busy schedule or exhaustion were discontinued at T2 and T3, respectively. Finally, 10 participants completed the program and all follow-up measures (Figure 1).

### 4.2. Sample Characteristics, Safety, Adherence, and Retention

Table 1 shows the sample characteristics. Among the four participants who reported having comorbidity, one had hypothyroidism and the other three had hypertension. None of the participants reported adverse events. The mean adherence rate of the TSHE was 61.5%. The adherence rate from the first through the fourth was 68%, 59%, 59%, and 60%, respectively. The mean adherence rate of the BAE was 66.3%. The adherence rate from the first through the fourth was 67%, 65%, 65%, and 68%, respectively. The retention rate was 76.9% for this sample.

### 4.3. Participant Experience

Table 2 lists quotations from participants by areas requiring attention for the design and implementation of a feasibility study. Participants met 50–60% of the exercise prescription and experienced the benefits of the intervention. This increased their motivation and confidence to continue performing the exercises. Moreover, participants expressed their willingness to continue this program on their own beyond the initial four weeks because it is easy to follow and convenient to do regardless of time or place. However, some participants failed to achieve the exercise prescription because of time constraints. In addition, the complexity of the intervention also hampered their reaching of their exercise goal. Therefore, they subjectively felt that the severity of their CIPN symptoms was only mildly reduced. 

### 4.4. Clinician-Assessed and Patient-Reported CIPN

Table 3 shows the change in CIPN from T1 to T5. The TNSc score increased from T1 to T2 and reached its peak at T2, then decreased over time from T3 to T4 and reached its lowest score at T5. Similarly, the NTX-13 score decreased from T1 to T2 and reached its lowest score at T2. It then increased gradually over time from T2, T3, to T4 and reached the highest score at T5. Contrary to the other measures, the ID-pain score decreased from T1 to T5. Moreover, the pain VAS decreased from T1 to T2 and mildly increased at T3, then decreased from T4 to T5. The pattern of change in research variables during the intervention period is shown in Figure 2. Table 4 examines the preliminary effectiveness of the intervention on CIPN. No significant difference was detected on clinician-assessed CIPN. However, a significant difference was found in patient-reported CIPN at T4 and T5.

## 5. Discussion

The enrollment rate, retention rate and prevalence rate of CIPN were 64.5%, 76.9%, and 14.8% in this study, respectively. The small sample size in this study may be due to the two inclusion criteria: (1) after completion of chemotherapy and (2) TNSc > 0. Eighty-nine participants subjectively reported CIPN symptoms but only 14.8% of survivors met inclusion criteria of TNSc > 0. The majority (84.3%) were not eligible. The discrepancy between subjective and objective CIPN measures were also found in the literature [30]. Moreover, results of a systematic review reported that 30% of breast cancer survivors suffered from CIPN after chemotherapy completion [31].

None of our participants reported adverse events. The mean adherence rates of this program were 61.5% for the TSHE and 66.3% for the BAE. Patient-reported CIPN significantly improved after the intervention. However, clinician-assessed CIPN was insignificant after the intervention. Participants reported that the program was effective and feasible. Moreover, they were willing to continue the program following study completion because it is easy to follow and convenient to do.

The enrollment rate of our study was 64.5%, which was higher than the results of an integrative review of the effects of exercise on CIPN, which indicated an overall enrollment rate of 53.16% [32]. Still, it is a challenge to recruit long-term cancer survivors and, according to Ganz et al., there are numerous barriers to doing to, such as locating patients, lack of institutional commitment, and lack of patient interest or response [33]. The prevalence of CIPN in our sample (14.8%) was similar to the results of a systematic review (11~80%) that examined CIPN in early-stage breast cancer survivors after having completed neurotoxic chemotherapy for at least 12 months [5]. None of our participants reported adverse events, which is consistent with the results of an integrative review of the effects of exercise on CIPN and an RCT examining the effects of exercise on CIPN that concluded that exercise intervention is safe [32,34].

The average adherence rate to our program was 61.5% for the TSHE and 66.3% for the BAE, which is similar to the home-based part (66.7%) of a comprehensive exercise rehabilitation program (supervision part: 97.5%; overall program: 83.1%) [32]. Although a home-based, individual exercise program is feasible for cancer survivors, adherence and fidelity would be challenging [34]. The retention rate of our study was 67.9%, which is lower than that of an RCT (85.7% at week four) that examined the effects of an eight-week yoga program on CIPN in gynecological cancer survivors suffering from CIPN [35]. One possible reason might be that the yoga program consists of twice-weekly, in-person, supervised training in addition to five times per week of individual video yoga, while our extremity exercise program is solely home-based and individual. As noted by Hatlevoll et al. with regard to an RCT examining the effects of an exercise program on colorectal cancer survivors, adherence to supervised exercise programs is higher than to home-based ones [36].

Our outcome measure was insignificant for clinician-assessed CIPN but significant for patient-reported CIPN after the intervention. Our significant result of subjective CIPN (NTX-13) after the intervention was consistent with the yoga arm of an RCT study at week 4 that proposed that yoga may increase blood flow and oxygen supply to protect peripheral nerves, which is similar to our extremity exercise program aimed at increasing blood flow [35]. However, our results differed from an interventional study examining the effects of a multimodal exercise program on CIPN in mixed types of cancer, which showed that exercise significantly improved subjective and objective CIPN after an eight-week intervention [37]. The difference might be due to our shorter intervention period and to our exercise program being solely home-based and individual, while the program of McCrary et al. included home-based and supervised multimodal exercises [37]. Our results were consistent with a comprehensive integrative review that found statistically significant CIPN improvement during a home-based exercise program at three and six weeks in duration of balance training as well as aerobic and strength training [32]. Moreover, a review study and a few research studies that observed subjective and objective CIPN on cancer survivors found an inconsistency in the low correlation between the two measures [4,5,37]. This may explain our inconsistent results on patient-reported CIPN and clinician-assessed CIPN after the intervention.

Results of our study indicated that CIPN symptoms measured by TNSc and NTX−13 worsened at one week after the intervention. Berthelot et al. identified the negative Hawthorne effect, which is defined as research participants reporting increased symptoms or overexpressing symptom severity during or after an intervention [38]. Our participants may have exhibited a negative Hawthorne effect because after survivors agreed to join this study, they were told to observe their CIPN symptoms and very much expected relief of those symptoms after doing the extremity exercises, even though this might not occur until after the initial week of intervention. Additionally, our study results on the increased score of TNSc from baseline to one week after intervention were similar to those of the eight-week exercise study conducted by McCrary et al. that utilized the same instrument [37].

Participants reported that they found the extremity exercise program to be feasible, effective, and easy to follow. However, survivors still reported two weaknesses of the program: A lack of time to perform the extremity exercises and the inconvenience and complexity of the BAE. Based on the recommendation of Bowen et al. to design a pilot intervention, our extremity exercise program was acceptable, requested by survivors, and practical, although there was some debate on its implementation in terms of the inconvenience of utilizing information technology and the complexity of the program’s prescription [29].

### 5.1. Strengths and Weaknesses

To our knowledge, this is the first study to introduce an extremity exercise program to manage CIPN symptoms in breast cancer survivors suffering from CIPN after cancer treatment completion. Moreover, the study adds information regarding the management of PN symptoms through exercise intervention with subjective CIPN improvement after intervention. Although the clinician-assessed CIPN was insignificant, results of our participants’ experience confirmed the feasibility and effectiveness of our program. Our extremity exercise program also met the Oncology Nursing Society’s research priorities for 2019 to 2022, which emphasize developing interventions to manage symptoms and improve survivors’ outcomes [18].

The weaknesses of this feasibility study were the following. Firstly, the prevalence rate of CIPN was relatively low because we recruited survivors who had completed cancer treatment and who were suffering from CIPN. Secondly, our adherence rate of exercise prescription was also lower and the duration of the intervention was shorter than that of other exercise studies as well as of individual, home-based ones. This low adherence rate may have affected the negligible results of the clinician-assessed CIPN. Lastly, as only 10 participants completed this study, our results will require validation through further interventional studies using a larger sample size.

### 5.2. Limitations

Our study was limited by a small sample size, the brevity of the intervention, and the lack of a control group. Moreover, the adherence rate of 61.5% for the TSHE and 66.3% for the BAE must be taken into account as a possible cause of the negligible clinician-assessed CIPN after the intervention. Additionally, twelve of the study participants were regular exercisers, and this may be a confounding factor of the study results. Therefore, the feasibility reported by participants has to be modest, due to the small sample size and a shortened exercise period. The efficacy of this study might be lost in the long run or many patients might quit earlier and thus mean it is less feasible. Moreover, the CPIN might correlate inversely with exercise, and patients are more motivated to maintain the habit.

## 6. Conclusions

The extremity exercise program may improve patient-reported CIPN in breast cancer survivors suffering from CIPN after cancer treatment completion. Furthermore, according to participants’ experience, the program is feasible, effective, and practical. However, as only 10 participants completed our study, it is necessary to increase the sample size and prolong the intervention period to confirm the results of this study. We recommend that future studies increase their sample size, add a control group, extend their intervention time, and formulate strategies to strengthen participant adherence to the exercise prescription.

## Figures and Tables

**Figure 1 healthcare-10-00688-f001:**
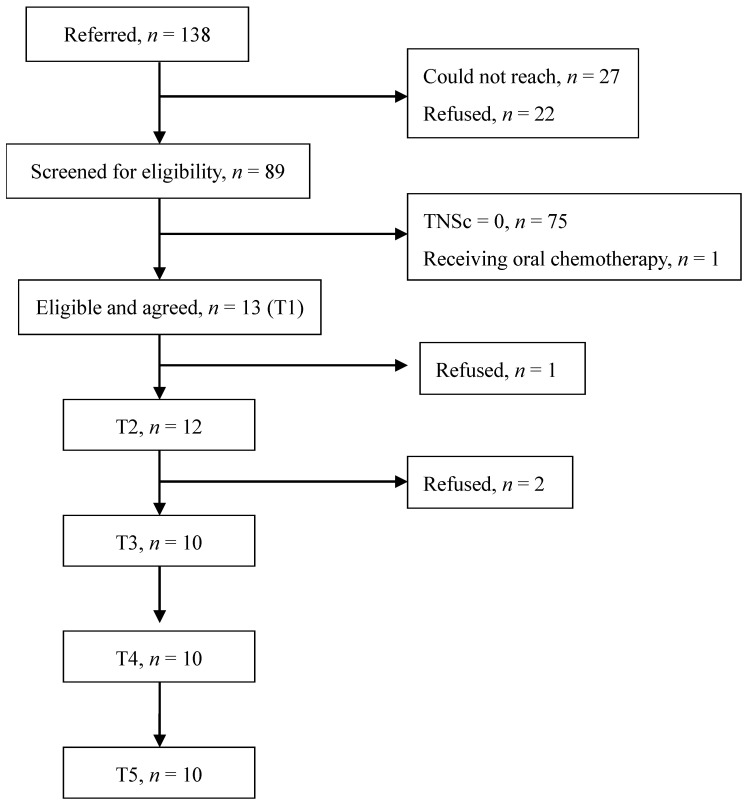
Participant recruitment and follow-up flowchart.

**Figure 2 healthcare-10-00688-f002:**
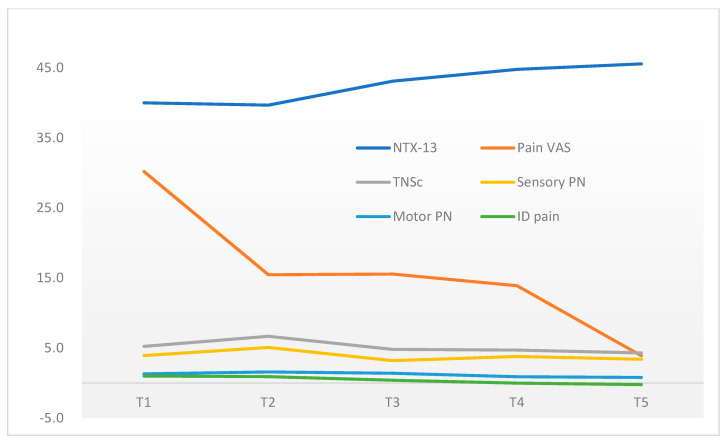
The change of CIPN over the study period.

**Table 1 healthcare-10-00688-t001:** Sample characteristics (*n* = 13).

Characteristics	Mean, SD	*n*	%
Age (42~65 years)	50.08, 5.79		
BMI (18.8~29.7 kg/m^2^)	24.78, 3.50		
Length since diagnosis (9.1~37.9 months)	24.11, 8.97		
Marital status			
Single		2	15.4
Married		11	84.6
Children			
No		2	15.4
Yes		11	84.6
Education level			
High school		2	15.4
College and above		11	84.6
Working status			
Not working		6	46.2
Full time		6	46.2
Part time		1	7.7
Household income			
Below average		11	84.6
Above average		2	14.4
Religion belief			
Non		1	7.7
Yes		12	92.3
Menopausal status			
Pre-		1	7.7
Post-		12	92.3
Comorbidity			
Non		9	69.2
Yes		4	30.8
Family history of cancer			
No		7	53.8
Yes		6	46.2
Regular exercise			
No		1	7.7
Yes		12	92.3
Experience of fall			
No		10	76.9
Yes		3	23.1
Cancer stage			
I		2	15.4
II		2	15.4
III		5	38.4
Unknown		4	30.8
Node status			
Negative		7	53.8
Positive		6	46.2
Gene-subtypes			
Triple negative		5	38.4
Her-2+, ER+		4	30.8
Her-2+, ER−		1	7.7
Her-2−, ER+		3	23.1
Surgery			
BCS		8	61.5
Total mastectomy		5	38.5

BMI: body mass index, Her-2: human epidermal growth receptor 2, ER: estrogen receptor, BCS: breast conserving surgery (The average yearly household income in Taiwan is NTD 1,200,000 in 2018).

**Table 2 healthcare-10-00688-t002:** Quotations from participants by areas needed to address the design and implantation of a feasibility study.

Aspects	Quotations from Participant’s Feedback
Acceptability	“This exercise program is easy to perform. I feel confident to do it.” “I believe that I will keep doing this exercise even after the four-week intervention. It’s really helpful that I even want to do it more.” “I feel comfortable and relaxed when doing Ten Skilled Hand Exercise outdoors or during commuting to work.”
Demand	“This is what I really needed to ease my CIPN symptoms.” “It is easy for me to do and does relieve my pain in extremity.” “The Ten Skilled Hand Exercise is exactly suitable for me to manage my numb fingers.”
Implementation	“It is hard for me to achieve the exercise goal because I had to work on weekdays.” “I have to sit on the bar table to do Burger Allen Exercise because my bed is too low to do this. It is inconvenient and complicate to do Burger Allen Exercise because I have to move from bed to bar table and back to bed.” “Due to the complexity of this exercise program, I have to do it with an iPad or on a television.”
Practicality	“After doing this exercise for one week, I feel my CIPN symptoms improved.” “When I do this extremity exercise program, I can feel the blood flow is going to the extremities. After doing the program, I can really feel the CIPN symptoms improved.” “I can feel the numbness is improving, thanks to this extremity exercise program. It is practical and takes no effort to perform.” “I do really experience the benefit of the exercise and want to do more Ten Skilled Hand Exercise and some more movements in toes of Burger Allen exercise during commuting to work or home.”

**Table 3 healthcare-10-00688-t003:** The change of CIPN from T1 to T5.

Measure	T1 (*n* = 13)	T2 (*n* = 12)	T3 (*n* = 10)	T4 (*n* = 10)	T5 (*n* = 10)
Variable Score	Mean, SD	Mean, SD	Mean, SD	Mean, SD	Mean, SD
TNSc	5.23, 3.06	6.67, 2.23	4.80, 2.78	4.70, 3.77	4.30, 3.65
Sensory CIPN	3.92, 2.87	5.08, 1.44	3.20, 1.93	3.80, 3.05	3.40, 3.13
Motor CIPN	1.31, 1.25	1.58, 1.38	1.40, 1.26	0.90, 0.99	0.80, 1.03
Autonomic CIPN	0, 0	0, 0	0.20, 0.63	0, 0	1.00, 0.32
Sensory symptoms	0.54, 0.78	0.92, 1.08	0.10, 0.32	0.30, 0.68	0.10, 0.32
Motor symptoms	0, 0	0.08, 0.29	0.20, 0.42	0.11, 0.33	0.10, 0.32
Pin sensitivity	2.08, 1.50	2.58, 1.38	1.50, 1.51	2.00, 1.89	1.90, 1.91
Vibration sensitivity	1.31, 1.55	1.58, 1.44	1.60, 1.71	1.50, 1.65	1.40, 1.71
Muscle strength	0.23, 0.60	0.17, 0.39	0.20, 0.42	0.10, 0.32	0, 0
DTR	1.08, 1.19	1.33, 1.30	1.00, 1.16	0.70, 0.95	0.70, 0.95
NTX-13	40.00, 6.58	39.67, 6.67	43.10, 8.60	44.78, 6.22	45.56, 6.31
ID-pain	1.00, 1.47	0.92, 1.38	0.40, 1.65	−0.10, 1.29	−0.22, 1.30
Pain VAS	30.19, 29.76	15.46, 29.45	15.56, 27.89	13.89, 24.47	3.89, 11.67

Abbreviation: SD: standard deviation; TNSc: Total Neuropathy Score- Clinical Version; CIPN: chemotherapy-induced peripheral neuropathy; DTR: deep tendon reflex; NTX-13: Functional Assessment of Cancer Therapy/Gynecologic Oncology Group- Neurotoxicity 13 Item Version; ID pain: Identification Pain; Pain VAS: Pain Visual Analogue Scale.

**Table 4 healthcare-10-00688-t004:** The change of CIPN severity over the study (*n* = 13).

	TNSc			Sensory CIPN			Motor CIPN			NTX−13			ID−pain			Pain VAS		
	B	SE	Wald *X^2^*	B	SE	Wald *X^2^*	B	SE	Wald *X^2^*	B	SE	Wald *X^2^*	B	SE	Wald *X^2^*	B	SE	Wald *X^2^*
Intercept	5.23	0.82	40.36 ***	3.92	0.68	33.61 ***	1.31	0.32	16.85 ***	40.00	1.75	43.44 ***	1.00	0.38	7.04 **	30.18	7.41	16.57 ***
T5	−0.93	1.25	0.56	−0.52	1.03	0.26	−0.51	0.48	1.10	4.50	1.28	12.31 ***	−1.22	0.59	4.30 *	−26.29	11.05	5.67 *
T4	−0.53	1.25	0.18	−0.12	1.03	0.01	−0.41	0.48	0.71	3.72	0.86	18.82 ***	−1.10	0.57	3.70	−16.29	11.05	2.17
T3	−0.43	1.25	0.12	−0.72	1.03	0.50	0.09	0.48	0.04	2.47	1.46	2.88	−0.60	0.57	1.10	−14.63	11.05	1.75
T2	1.44	1.19	1.46	1.17	0.98	1.41	0.28	0.46	0.36	−0.26	1.23	0.05	−0.08	0.54	0.02	−14.73	10.45	1.97
T1	0			0			0			0			0			0		

Reference group, T1 Note: * *p* < 0.05, ** *p* < 0.01; *** *p* < 0.001. Abbreviation: TNSc: Total Neuropathy Score- Clinical Version; CIPN: chemotherapy-induced peripheral neuropathy; NTX-13: Functional Assessment of Cancer Therapy/Gynecologic Oncology Group- Neurotoxicity 13 Item Version; ID pain: Identification Pain; Pain VAS: Pain Visual Analogue Scale; SE: Standard Error.

## Data Availability

Ya-Jung Wang keeps the research data and material.

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
