# Peer review of "Extremity Exercise Program in Breast Cancer Survivors Suffering from Chemotherapy-Induced Peripheral Neuropathy: A Feasibility Pilot Study"

_healthcare, 2022, doi:10.3390/healthcare10040688_

Round 1

Reviewer 1 Report

Dear authors, in the attached document you can find the considerations and suggestions for improving your work.
Best regards.

Reviewer 2 Report

The manuscript entitled "Extremity Exercise Program in Breast Cancer Survivors Suffering from Chemotherapy-induced Peripheral Neuropathy: A Feasibility Study" is a sound study. I appreciated the whole study design and the drawn conclusion. The only limit is the number of subjects that completed the program: it is below the power size threshold evaluated by the authors. 

Reviewer 3 Report

I think it is a good study. Well structured and developed. Original as introduce an extremity exercise program to manage PN symptoms in breast cancer survivors. The results, although good for the patient, which is what matters, do not seem good for clinical judgement. The number of subjects is low, and the absence of control group, even it is understood, are strength limitations as the authors explain, but that cannot be changed now. I consider it to be a very interesting pilot study that could be useful as a reference for setting up working models for patients with this pathology.
Minor changes:
In line 49 it is written "parathesia" shouldn't it be paresthesias?

Reviewer 4 Report

The paper deals with an extremity exercise program in order to treat chemotherapy-induced peripheral neuropathy (CPIN). The treatment of CPIN is an unmet need in medical oncology. However, I suggest to review some aspects of the paper. 

In the abstract, I would better specify the passage where the programme is defined as "effective" but at the same time it is written "...improved patient-reported PN .... but was insignificant on clinician-assessed PN".

In the introduction, please specify that BC is the first cancer in the female gender.

Consistently with literature data reported by the authors, and that the final number of patients completing the program (10) is lower the sample size suggested by literature (12), I suggest to extend the sample size at least reaching 12 patients completing the excercise program. More in general, probably more patients and a longer follow-up/exercise period are needed to evaluate the real feasibility (the efficacy might be lost in the long run and many patients might quit earlier and thus be less feasible or maybe CPIN might correlate inversely with exercise and patients are more motivated to maintain the habit). I would discuss this within the limitations of the study.

It might be useful to assess and investigate the impact of the program on quality of life and daily activities. 

Of 13 patients, 12 were already doing regular exercise: this could be a confounding factor. Please, discuss this point among the limitations of the study.

It would be interesting to see what happens if patients do exercises during chemotherapy when they start to present symptoms or even before (in prevention), rather than after the damage on the nerve fibres has set in.

I suggest looking further into the concept "the conceptual model of increasing blood supply to peripheral nerves to reduce neuropathy symptoms". Indeed, there is no strong evidence on the efficacy of exercise in reducing symptoms, and more importantly it remains to be demonstrated why improved vascularisation/oxygenation of tissues would benefit already damaged nerve fibres. On the contrary, increased perfusion to a certain body district has often been associated with increased drug distribution in the district.

Reviewer 5 Report

The authors developed an interesting intervention to manage neuropathic pain in survivors of breast cancer; however, the quality of the research performed is low. 

Some comments of the different sections of the manuscript are detailed below: 

Abstract: 

1) Line 34, you use the abbreviation PN when before you have used CIPN. Please confirm through all manuscript that you use the same abbreviations.

Please check within the reccomendations of the journal if a conclusion is needed on the abstract. 

Introduction: 

*The statement "breast cancer is the most common cancer" does not refer to anywhere. Even if it is worldwide, you could include some information about the newly diagnosed that received chemotherapy worldwide and then about Taiwan. 

*Line 43: newly diagnosed breast cancer survivors does not make sense. I would delete survivors, as at the moment they are diagnosed, they are still patients and suffering the illness.

*Line 45: similar to the previous comment, I would check the term survivors. Usually this is used when treatment is finished and there is a remision of the pathology. I would not use this term when they're receiving chemotherapy.

*Line 48: "cancer survivors who suffered from CIPN"; could you say it in present tense? as this is something they have to live with.

*Lines 53, 61 and 69: you say "neuropathy symptoms" and then after "neuropathic symptoms" please homogenize this.

*Line 78: add a reference if needed on the verification process of the TSHE effects. 

Materials and methods: 

*Line 92: I would add a comma after "five times" to specify when outcome were assessed

*Line 94: Please add (number) before each inclusion criteria and exclusion criteria

*Line 98: What about the Declaration of Helsinki? this needs to be included. 

*Sample size calculation: I would reccomend to delete the first sentence. This sample size between 8 and 1299 does not make sense. If you recruited a minimun of 12 based on the study of Sim, then mention only this. 

*Clinician-assessed PN: even if you have used CIPN, the abbreviation PN is not used before. Please homogenize these concepts. This happens again on the participant-reported PN

*Please include information about de Intra class correlation of the Visual Analogue Scale (and mention it before using the abbreviation)

Results: 

*Only 10 participants were included on the study. This is even less than your sample size calculation. In my opinion, the results you obtained are not relevant enough 

*This sample size does not match with what you state on your introduction about the breast cancer being the most prevalent cancer worldwide, and when neuropathic symptoms are quite frequent on this population

*Figure is difficult to understand 

*Table 3 does not include information about the p values between assessment points. 

Discussion

*Line 282- you compare your study with and interventional study of yoga, home-based exercise, but you don't specify which type of study are these. If these are not similar, then it is dificult to compare results. 

*In my opinion, the limitations you expose are heavy enough for your study to not to be published when it is about a study of feasibility 

Round 2

Reviewer 4 Report

The authors have given clear answers and their corrections make the publication fit for publication after further, minor revisions. Considering that the minimum sample size was not reached, I suggest specifing the reasons of the difficult accrual in the study and I would modify the text and title by specifying that this is a pilot study.

Reviewer 5 Report

Authors have correctly followed the suggestions done to improve their study. In my opinion, even if the sample size and the results continue being too small, it could be published as a Feasibiliy Pilot Study. I would thus suggest to change this into the title of the manuscript and include the word "pilot". 
